# Adaptive responses of histone modifications to resistance exercise in human skeletal muscle

Changhyun Lim[1], Junya Shimizu[2], Fuminori Kawano[2,3]*, Hyo Jeong Kim[4], Chang Keun Kim[5,6]

1 Department of Kinesiology, McMaster University, Ontario, Canada, 2 Department of Sports and Health Science, Matsumoto University, Nagano, Japan, 3 Graduate School of Health Sciences, Matsumoto University, Nagano, Japan, 4 Department of Healthy Ageing, Korea National Sport University, Seoul, Korea, 5 Exercise and Metabolism Research Center, Zhejiang Normal University, Jinhua, China, 6 Human Physiology, Korea National Sport University, Seoul, Korea

⊛ These authors contributed equally to this work.
* kawano@t.matsu.ac.jp

**Data Availability Statement:** Raw data for RNA-seq are available in the National Center for Biotechnology Information (NCBI) Bio Project (ID: PRJNA560308) database. Summary of full data

## Abstract

Exercise training causes epigenetic changes in skeletal muscle, although it is unclear how resistance exercise (RE) affects histone modifications. The present study was carried out to investigate the effects of acute RE and RE training on gene expression profiles and histone modifications in human skeletal muscle. Healthy male adults were assigned to acute RE (n = 9, age = 20.5±4.3yr, BMI = 28.0±6.8kg/m$^2$) or RE training (n = 21, age = 23.7±2.5yr, BMI = 24.2±2.7kg/m$^2$) groups. Biopsy samples were obtained from the vastus lateralis muscle before and three hours after a single bout of acute RE, or 3-days after 10 weeks of RE training. RNA sequencing analysis revealed that 153 genes with GO terms including muscle development, stress response, metabolism, cell death, and transcription factor were significantly up-regulated (+291% vs. pre-acute RE) upon acute RE. Expressions of these genes were also greater (+9.6% vs. pre-RE training, p<0.05) in RE trained subjects. Significant up-regulation of acetylated histone 3 (H3) (+235%) and H3 mono-methylated at lysine 4 (+290%) and tri-methylated at lysine 27 (+849%), whereas down-regulation of H3.3 variant (−39%) distributions relative to total H3 were observed at transcriptionally activated loci after acute RE compared to pre-acute RE levels. Interestingly, the distribution of acetylated H3 was found to be up-regulated as compared to the level of total H3 after RE training (+40%, p<0.05). These results indicate that a single bout of RE drastically alters both gene expressions and histone modifications in human skeletal muscle. It is also suggested that enhanced histone acetylation is closely related to up-regulation of gene expressions after RE training.

## Introduction

Gain of exercise-induced effects on skeletal muscles differs between individuals. Bamman et al. [1] reported that subjects were classified into extreme responders, modest responders, and

obtained from RNA-seq is also available in Supporting Information.

**Funding:** The present study was supported by Japan Society for the Promotion of Science KAKENHI Grant-in-Aid 16H03263 and 18H04987 to F. Kawano.

**Competing interests:** The authors have declared that no competing interests exist.

non-responders depending on the magnitude of muscle fiber hypertrophy induced by resistance exercise (RE) training. It was also reported that the up-regulation of mechano-growth factor and myogenin gene expression was heightened in both extreme and modest responders after RE training. Previous studies [2, 3] further reported muscle mass gain responses after RE training are positively correlated with the expression levels of a particular microRNA, miR-378. Therefore, we hypothesized that epigenetic regulation causes the differences in the responsiveness of skeletal muscle to exercise training.

Histone modification is regarded as one of the epigenetic systems that plays a crucial role in the transcriptional activity in the cell nucleus. Slow- and fast-twitch skeletal muscles have different histone modification patterns. Transcriptionally active histone modifications, such as acetylation and tri-methylation at lysine 4 of histone 3 (H3), were found to be prevalent at the loci with higher expression in fast-twitch muscles of adult rats, although no relationship between histone modifications and gene expression was seen in slow-twitch muscles [4]. Masuzawa et al. [5] also reported that in response to the acute running exercise using a treadmill, the transcriptional activation of peroxisome proliferator-activated receptor gamma coactivator 1-alpha gene in rats was greater in the fast-twitch muscles in which the acetylation of histones was more prevalent as compared to that in the slow-twitch muscles. Begue et al. [6] demonstrated that fiber type-specific DNA methylation in human skeletal muscle, showing CpG sites of genes selectively expressed in type 1 or IIa myosin heavy chain fibers were hypomethylated. These data suggest that epigenetic regulation based on muscle fiber type characteristics affected the responsiveness of genes to exercise.

We further reported that endurance exercise training decreased the responsiveness of genes to muscular unloading in association with enhanced incorporation of histone variant H3.3 into the nucleosomes in the plantaris muscle of rats. This indicates that endurance exercise training stimulated the turnover of histones [7], but it is currently unclear how RE affects the distribution of histones and their modification patterns in animal models. Furthermore, even less research has been conducted on the epigenetic response to RE in human skeletal muscle [8–11]. Therefore, the present study is built upon the animal model literature, and aims to investigate the response of gene expression profiles and histone modifications in human muscle before and after acute RE and chronic RE training, respectively.

## Materials and methods

### Experimental design and ethical approval

In the present study, we analyzed skeletal muscle biopsy samples obtained after two different experiments, acute RE and RE trainings. Both experiments were approved by the Sport Ethics Committees of Hochiminh City University (33/QD-TDTTHCM-SDH), and the Bioethics Committee of Korea National Sport University (1263-201706-BR-002-01), respectively. Changhyun Lim, Hyo Jeong Kim, and Chang Keun Kim performed the experiments and sample collections for acute RE experiment in Hochiminh City University, Vietnam, and for RE training experiment in Korea National Sport University, Korea. Since their affiliation was with Korea National Sport University when they performed these experiments, the research proposal was approved by the Bioethics Committee of Korea National Sport University for RE training experiment. The acute RE experiment was a collaboration study with Hochiminh City University, therefore the ethical approval was obtained in Hochiminh City University. Original data obtained in these studies have been published [12, 13]. We combined the samples obtained from these studies to analyze for additional data shown in the present study. The original studies were also carried out in compliance with the Declaration of Helsinki. All subjects were informed about the purpose of the study, the experimental procedures, possible

risks and discomforts they might experience during experiments, and written informed consent from all subjects were obtained. We analyzed human samples in a fully anonymized and randomized manner, only Changhyun Lim, Hyo Jeong Kim, and Chang Keun Kim had access to subjects' information. Although the present study reports the results for gene expression patterns and epigenetic changes upon acute RE and RE training experiments, a portion of the results has been published [12 and 13]. Therefore, detailed information of subjects and exercise protocols are briefly mentioned in the following sections.

## Experiment 1

**Subjects.**  Nine male weight lifters (age = 20.5±4.3yr, BMI = 28.0±6.8kg/m$^2$, mean ± SD) participated in Experiment 1 (acute RE). All subjects were weightlifters of national caliber of H City team including a London Olympic medalist. They have been in the same training for at least seven years, and shared similar living condition for diet, nutrition, and dormitory.

**Exercise protocol.**  A week prior to the test, subjects completed a 10-repeated maximum (RM) squat and bench press test to ensure appropriate exercise intensity (172.2 ± 38 kg, 82.6 ± 16.4 kg, 1RM of squat and bench press, mean ± SD, respectively). Intensity of exercise was then set at 60% of their 1RM weight and subjects completed three sets of 6-repetitions during each session of exercise. This RE consisted of squat, single leg lunge, and deadlift, and was repeated twice by all subjects.

**Collection of muscle biopsy samples.**  Muscle biopsy samples were obtained using local anesthesia (1% lidocaine) administrated into the mid belly of the vastus lateralis muscle immediately before (pre-acute RE), and three hours after (post-acute RE) exercise. Muscle biopsy samples were frozen in liquid nitrogen and stored at −80˚C until further analysis. Nine biopsy samples were combined to analyze for gene expressions in Experiment 1. Further, three out of nine biopsies were selected for the histone modifications analysis. Throughout Experiment 1, biopsies of pre- and post-acute RE were selected from the same subjects.

## Experiment 2

**Subjects.**  Effects of RE training were examined in 21 males (age = 23.7±2.5yr, BMI = 24.2 ±2.7kg/m$^2$) who had no record of medical disorders in the musculoskeletal, cardiovascular, and respiratory systems and had not undergone any regular resistance exercise in the last two years. The subjects were separated into three groups; 80FAIL (n = 7, age = 24.5±1.8yr, BMI = 25.9±3.9kg/m$^2$), 30WM (n = 7, age = 23.1±2.0yr, BMI = 25.0±3.1kg/m$^2$), and 30FAIL (n = 7, age = 23.0±1.2yr, BMI = 24.4±1.3kg/m$^2$).

**Exercise protocol.**  In Experiment 2, all subjects performed three repeated sets of leg press, leg extension, and leg curl, three times per week for 10 weeks. Considering the risk of injury to subjects who had not performed any resistance exercise before, 1RM was determined by the indirect measurement method suggested in a previous report [14]. The 80FAIL group exercised at 80% of 1RM until they could not achieve muscular contractions in every set, the 30WM group performed the total workload matching that of the 80FAIL, and the 30FAIL group exercised at 30% of 1RM until failure. For example, total work volume performed in leg press was 2,358–3,030; 2,495–3,060; and 2,995–3,588 kg/set in 80FAIL, 30WM, and 30FAIL, respectively [12].

**Collection of muscle biopsy samples.**  Muscle biopsy was sampled from the mid belly of the vastus lateralis muscle by aforementioned procedures prior to training (pre-RE training), and 72 hours after the final session of RE (post-RE training). Our earlier study has reported that there was significant difference in the muscle fiber size by time (pre vs. post) in all groups by two-way ANOVA [12]. Therefore, in the present study nine biopsy samples from

individuals subjected to RE training were randomly collected from the three groups (21 subjects) and were analyzed for gene expressions. Further, three out of nine biopsies were selected for histone modifications analysis. Biopsies of pre- and post-RE training were selected from the same subjects in Experiment 2.

## RNA sequencing (RNA-seq)

Pieces of frozen muscle samples for each group (~100 mg in total) were pooled and homogenized in 1 mL ISOGEN (NIPPON GENE, Tokyo, Japan). RNA extraction was performed according to manufacturer's instructions. RNA-seq analysis was performed through commercial service (Novogene, Chula Vista, CA). mRNAs were enriched with oligo(dT) beads and randomly fragmented in the fragmentation buffer, followed by cDNA synthesis using random hexamers and reverse transcriptase. After the first-strand synthesis, a custom second-strand synthesis buffer (Illumina, San Diego, CA) was added along with dNTPs, RNase H, and Escherichia coli polymerase I to generate the second strand by nick translation. The final cDNA library was prepared after purification, terminal repair, A-tailing, ligation of sequencing adapters, size selection, and PCR enrichment. The HiSeq-PE150 system (Illumina) was used to obtain reads of 150-bp paired ends. Approximately 50 million reads for each group were mapped to the human whole genome database using the TopHat2 software, and the fragments per kilobase of exon per million mapped fragments (FPKM) value was calculated for the exons of all known loci. Raw data for RNA-seq are available in the National Center for Biotechnology Information (NCBI) Bio Project (ID: PRJNA560308) database.

Gene ontology (GO) analysis was performed for genes that were up- or down-regulated with more than 2-fold differences when compared between the pre- and post-acute RE samples. Genes encoding for mitochondrial DNA were excluded from the GO analysis, because mitochondrial DNA does not form chromatin structure that is essential for subsequent analysis of histone distributions. Furthermore, the up-regulated genes annotating for at least three out of five major GO terms, stress response, muscle development, cell death, metabolic process, and transcription factor, were targeted for histone distributions analysis.

## Chromatin extraction

Chromatin extraction was performed as described previously [4]. Three randomly chosen muscle samples from the same subjects between the pre- and post-experiment of each group were pooled. The pooled muscle samples (~80 mg) were homogenized in cooled PBS. After centrifugation at 12,000 × g, the pellet was fixed in 1% paraformaldehyde on ice for 10 min followed by quenching in 200 mM glycine. The pellet was then resuspended in lysis buffer (50 mM Tris-HCl, 1% SDS, and 10 mM EDTA, pH 8.0) and sonicated using Sonifier 250 (Branson, Swedesboro, NJ). In order to obtain an average DNA fragment size of 500 bp, 24 s of continuous sonication followed by 30 s cooling on ice was repeated four times. After centrifugation at 12,000 × g, the supernatant was stored as the chromatin-rich extract at −80˚C until further analysis.

## Chromatin immunoprecipitation (ChIP)

ChIP was also performed as described previously [4]. Chromatin samples extracted from each group were diluted to contained around 700 ng DNA. Chromatin was incubated overnight at 4˚C with antibodies diluted at 1:50 for: anti-H3.3 (ab176840, Abcam, Cambridge, UK), anti-pan-acetyl H3 (39139, Active Motif, Carlsbad, CA), anti-H3 mono-methylated at lysine 4 (H3K4me1, 5326S, Cell Signaling Technology, Danvers, MA), anti-H3 tri-methylated at lysine 27 (H3K27me3, 9733S, Cell Signaling Technology), or anti-total H3 (4620S, Cell Signaling Technology), followed by subsequent reaction with protein G agarose beads (9007, Cell

Signaling Technology; 20 µl for each reaction) for 4 h at 4˚C. Beads were washed and incubated with proteinase K (Takara Bio, Shiga, Japan) for 1 h at 65˚C. DNA was extracted by adding phenol-chloroform solution (25:24) and by centrifugation at 12,000 × g for 10 min at 20˚C. The supernatant was collected, and ethanol precipitation was performed using Ethachinmate (NIPPON GENE). The final pellet was resuspended in tris-EDTA buffer and stored at −20˚C. The amount of input DNA in the chromatin utilized for ChIP reaction was estimated with the same procedure but without any antibodies.

## Quantitative PCR (qPCR)

In order to quantify the distribution of histones at the target loci, qPCR was performed using StepOne Real Time PCR System (Thermo Fisher Scientific, Waltham, MA). The THUNDER-BIRD qPCR Mix (TOYOBO, Osaka, Japan) was used for the PCR reaction according to manufacturer-recommended dilution procedures. Primer pairs of 1 kbp downstream from the transcription start site (TSS) of 16 target genes were designed (Table 1). Quantification of qPCR results was performed by normalizing cycle-to-threshold (Ct) of the target amplification with Ct of the respective input DNA (% input). Results were normalized further using the median within each gene, and averaged for the 16 target genes.

## Western blotting

Three muscle samples were randomly selected from each group to analyze for protein expression levels. Total histone was extracted using the Epiquik Total Histone Extraction Kit (Epigentek, Farmingdale, NY). Total histone obtained from the 25 mg muscle sample from each group was extracted in 500 µL lysis buffer packaged in the kit, centrifuged at 12,000 × g for 5 min at 4˚C, of which 300 µL supernatant was collected and mixed with 90 µL balance buffer packaged in the kit. The total histone extract was further dissolved in an equal amount of 2X SDS sample buffer (20% glycerol, 12% 2-mercaptoethanol, 4% sodium dodecyl sulfate, 100 mM tris-HCl, and 0.05% bromophenol blue, pH 6.7). Western blotting was carried out as described previously [4]. Antibodies specific to H3.3 (ab176840, Abcam), pan-acetyl H3

**Table 1. List of target genes and sequences of primer pairs for ChIP-qPCR analysis.**

| Gene symbol | Forward primer | Reverse primer |
|---|---|---|
| Ankrd1 | TTACTTCGGTTCCCAGGTTG | CAGCTTGGTGATTTGGAGGT |
| Atf3 | CCTTGACATTCCTGCCTGTT | CTCCAGGGCTTTTCCTCTCT |
| Btg2 | ACAATTTGGAGTCCCAGTGC | CGGGCTGCTTATCTCTTCAC |
| Cryab | AGTGAGAGCAACGAGGGTGT | ACCGTTTGTGAGGGTCTCAG |
| Csrp3 | GCTAGCATTGAGGACCCAAA | GCCTCCATCCCTAACCTTTC |
| Dnajb | ACAAACACACGCTTGCACTC | CCACCTCCTTGGACTCTCAG |
| Fos | GGGACGCTCCAGTAGATGAG | AGTGCAGACCAGAGGTTGCT |
| Hspa1b | TGGTGCTGACCAAGATGAAG | CCCAGGTCAAAGATGAGCAC |
| Junb | TGGAACAGCCCTTCTACCAC | GAAGAGGCGAGCTTGAGAGA |
| Lmcd1 | CACGCACGCAGTTTCTTTAG | TCCTGTGCAGGAGTTTACCC |
| Myf6 | AGAGAAAATCTGCCCCCACT | GCATCTTCTCCTGCTGATCC |
| Socs3 | ATTCGGGACCAGGTAGGAAG | GTGTGGACGGAGGGAGAAAC |
| Ubc | ATCGCTGTGATCGTCACTTG | CCACCTTGTTTCAACGACCT |
| Vegfa | TCCGGGTTTTATCCCTCTTC | ACCCCGTCTCTCTCTTCCTC |
| Vgll2 | CCACCAGGTACGTGTCTCCT | TAGCAGGGCTTAGCTGCTTC |
| Zep36 | CAGCTTGGTGATTTGGAGGT | CTGAGACTTCAGCCCCAGAG |

(39139, Active Motif), H3K4me1 (5326S, Cell Signaling Technology), H3K27me3 (9733S, Cell Signaling Technology), or total H3 (4620S, Cell Signaling Technology) were used to detect each protein. Antibody-bound protein was detected using a chemiluminescence based method using the Western BLoT Hyper HRP Substrate (Takara Bio). Quantification of bands was performed using the image analyzing software (Image J). Protein levels were quantified based on the integrated density of the band, which was calculated as the mean density multiplied by the band area. The values of band intensity were further normalized with respective total H3 levels.

## Statistical analysis

Statistical analysis was performed using BellCurve for Excel (Social Survey Research Information Co., Ltd.). For the data analysis of RNA-seq, all FPKM values were compared for the same genes between all experimental groups, and a correlation coefficient (R2) was calculated (Fig 1). FPKM values were normalized using the median within each gene, and averaged for the up- or down-regulated genes (Fig 2). For ChIP data, a boxplot was used to display the distribution of the data obtained from each gene (Fig 5 and 6). Values plotted in the Figs 2, 5, 6, and 7 were compared to determine the significant differences only for between pre- and post of acute RE or RE training. Because the data of pre- and post-groups were obtained from the same subjects in both Experiment 1 and 2, significant differences were examined using a paired t-test. Differences were considered significant at $p < 0.05$.

## Results

### Gene expression

Fig 1 shows the correlation of gene expression among all experimental groups. Relatively lower correlation was observed between pre- and post-acute RE (R2 = 0.967) as compared to that between pre- and post-RE training (R2 = 0.987). In Experiment 1, expression levels were significantly up-regulated (+291% vs. pre-acute RE, Fig 2A) for 153 genes, and down-regulated (−72% vs. pre-acute RE, Fig 2B) for 29 genes. However, only nine genes were significantly up- (4 genes) or down- (5 genes) regulated in the RE training groups of Experiment 2. The complete list of genes and FPKM values are available in the supporting information on the journal web site (S1–S3 Files). Results of GO analysis showed that the major GO terms including genes up-regulated upon acute RE, were muscle development, stress response, metabolism, cell death, and transcription factor (Fig 3). Terms for down-regulated genes included collagen, and extracellular matrix (Fig 4). The levels of gene expressions were analyzed for the same gene sets in both Experiment 1 and 2. Expression levels of the up-regulated genes were significantly greater (+9.6%) in post-RE training than that in pre-RE training, although no changes were observed after RE training in the down-regulated genes (Fig 5).

### Distribution of histones

In the present study we analyzed the distribution of histones at the loci of 16 targeted genes that were up-regulated upon acute RE and related to more GO terms in biological processes,. These genes were selected based on two requirements: genes that showed high FPKM values and bore the downstream regions of TSS that could be amplified using PCR; and genes that conformed to at least three major GO terms like stress response, muscle development, cell death, metabolic process, and transcription factor.

**Histone variant H3.3.** The distribution of H3.3 was decreased (−86% and −39% in absolute and relative levels to total H3 vs. pre-acute RE, respectively; $p < 0.05$) in post-acute RE

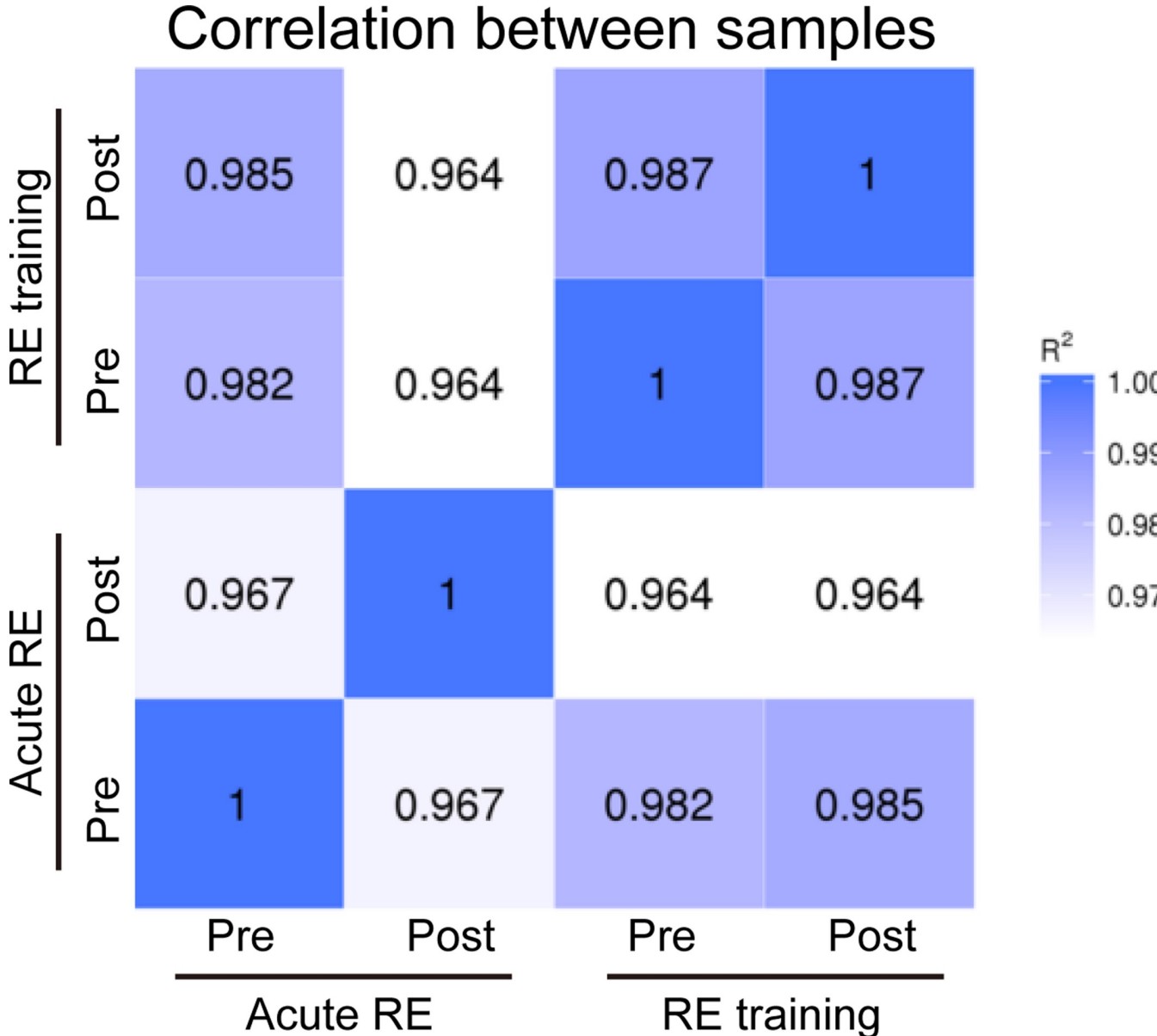

**Fig 1. Correlation of gene expression among all experimental groups.** Expressions of all genes identified by RNA sequencing were compared. Numbers shown in squares indicate the coefficient of each comparison. Denser color indicates higher correlation in the gene expression profiles of samples.

(Fig 6A and 6F). Absolute H3.3 distribution was also lowered after RE training (−25% vs. pre-training), however, levels of H3.3 distribution were similar to the pre-RE training group levels when values were normalized using total H3 distribution (Fig 7A and 7F).

**Acetylated H3.** Considering the absolute level of acetylated H3, the distribution of pan-acetyl H3 was unchanged in response to both acute RE and RE trainings (Figs 6B and 7B). However, the relative distribution of total H3 was significantly greater after acute RE and RE trainings (+235% and +40% vs. respective pre-groups, Figs 6G and 7G).

**H3K4me1.** The absolute level of H3K4me1 distribution was unchanged upon acute RE, although the relative distribution was significantly increased (+290% vs. pre-acute RE) in response to acute RE (Fig 6C and 6H). Absolute level of H3K4me1 distribution was also

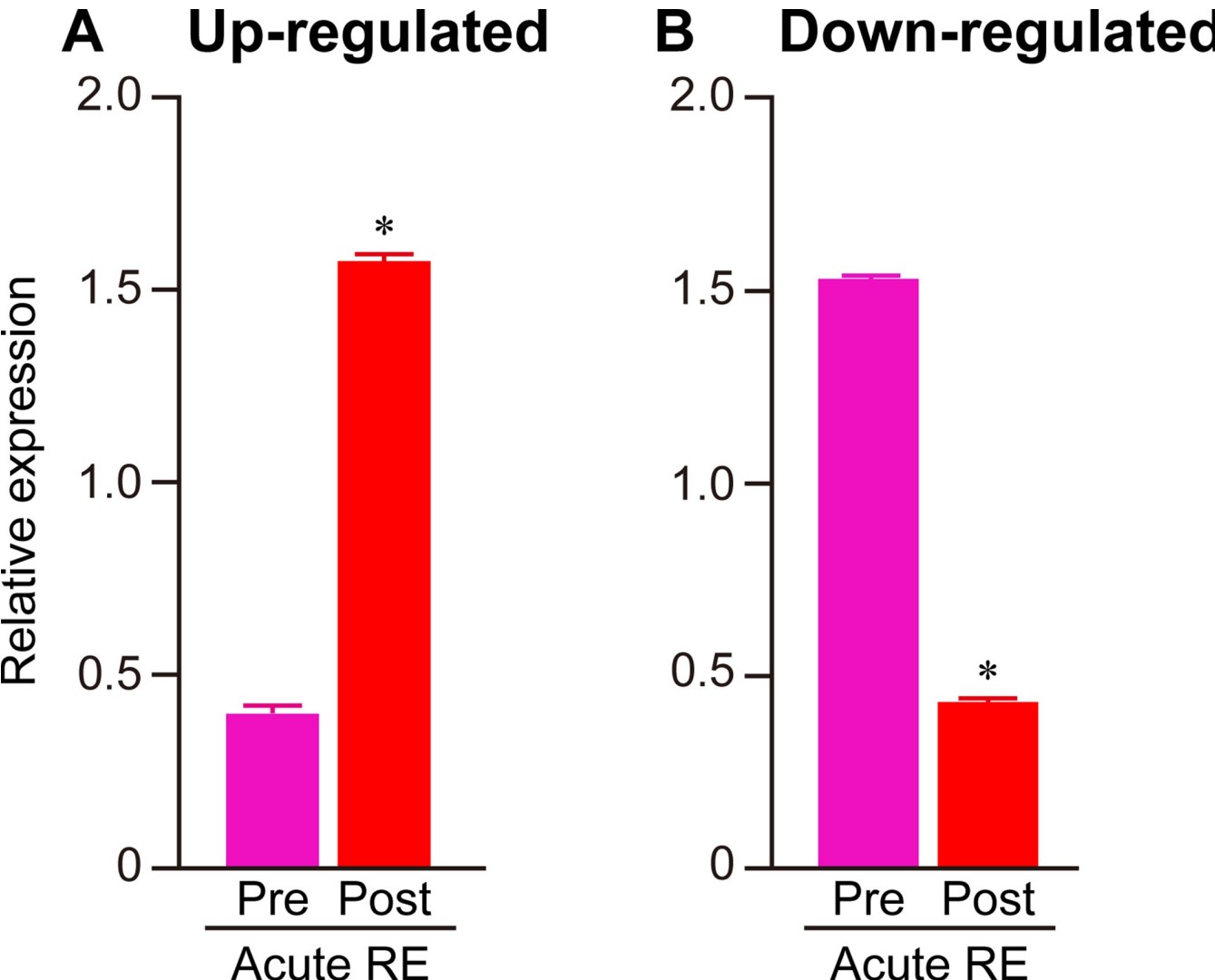

**Fig 2. Responses of gene expressions to acute RE.** Quantitative analysis of 153 up-regulated (A) and 29 down-regulated (B) genes before (Pre) and after (Post) acute RE. FPKM values obtained from RNA-seq were used to calculate the mean expression levels in both groups. To normalize the differences in the distribution of data among the genes, the median value was calculated within each gene and expressed as 1. Mean ± SE. *: $p < 0.05$ vs. respective pre group, examined by paired $t$-test.

increased (−39% vs. pre-training, $p < 0.05$) after RE training (Fig 7C). However, it appeared to decrease (−23% vs. pre-training) after RE training when values were normalized using total H3 distribution ($p = 0.05$, Fig 7H).

**H3K27me3.** A drastic increase in H3K27me3 distribution was observed after acute RE at target loci (+153% and +849% in absolute and relative levels to total H3 vs. pre-acute RE, $p < 0.05$, respectively; Fig 6D and 6I). Although the absolute levels of H3K27me3 distribution were similar between pre- and post-RE training groups (Fig 7D), the relative distribution of H3K27me3 appeared to be greater (+54% vs. pre-training) after RE training ($p = 0.05$, Fig 7I).

**Total H3.** Levels of total H3 distribution were also altered upon stimulation induced by exercise (Figs 6E and 7E). H3 distribution was significantly decreased (−76% and −19% vs. respective pre-groups) after acute RE and RE training.

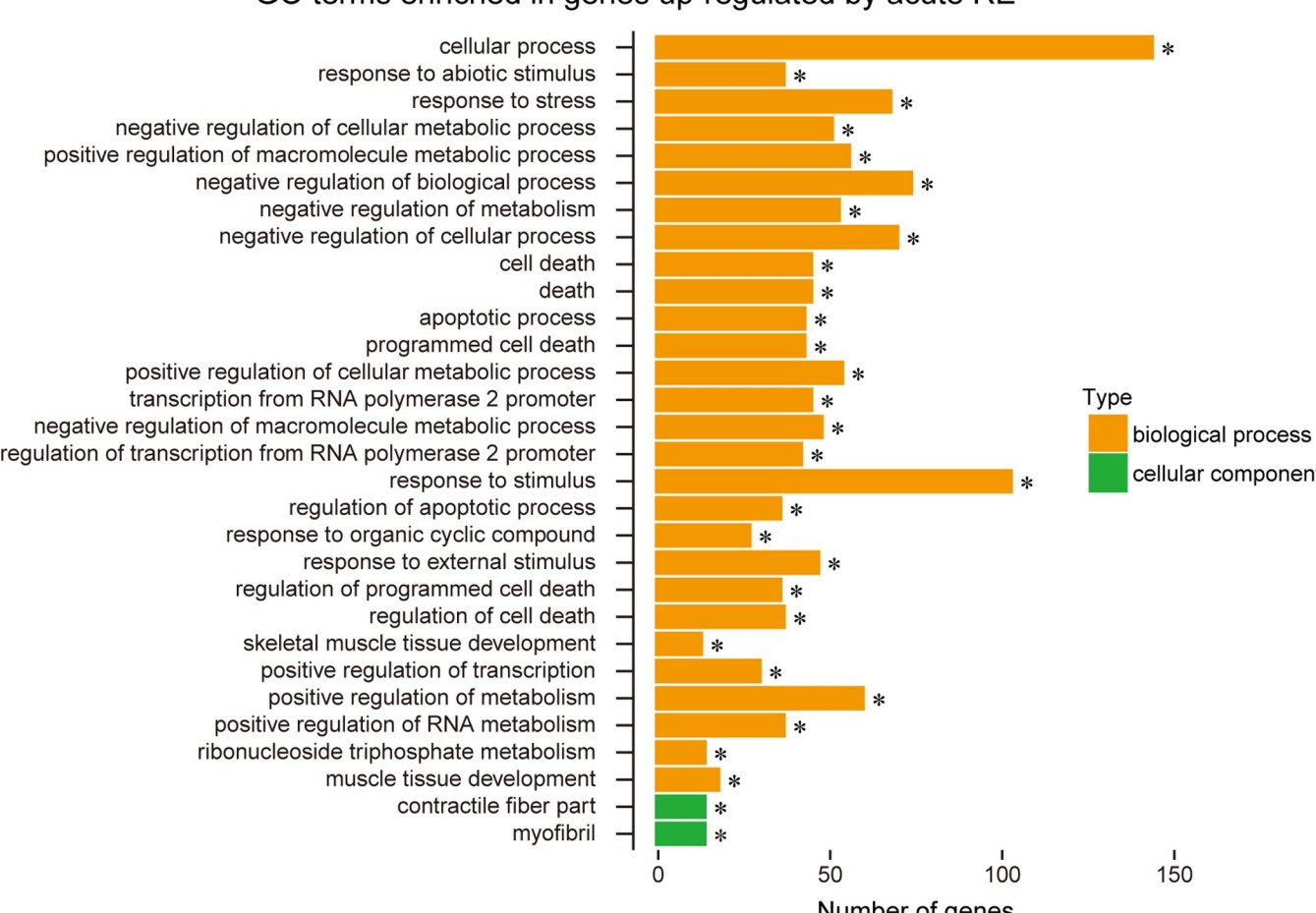

**Fig 3. Results of GO analysis in up-regulated genes.** 153 genes were targeted for analyzing their GO terms. Significant enrichment was found in 28 terms of biological process (orange), and 2 terms of cellular component (green). Note that the up-regulated genes frequently include the terms such as muscle development, stress response, metabolism, cell death, and transcription factor.

**Western blotting.** Levels of H3.3, pan-acetyl H3, H3K4me1, and H3K27me3 expression were unchanged upon acute RE or RE training when values were normalized using total H3 levels (Fig 8).

## Discussion

### Limitations of analysis

The present study combined human biopsy samples with the previous research group of which data for the histochemical characteristics and protein expression were published [12 and 13]. However, due to the limited volume of remaining samples, we could not analyze all the samples individually. Therefore, nine biopsies were randomly chosen in Experiment 2, and three were selected for ChIP and western blot analysis in both experiments. The biopsies of pre- and post-groups in both Experiment 1 and 2 were obtained from the same subjects.

Epigenetic regulation differs between muscle fiber types [6]. For consistency in analyzing data, in the present study histone modifications should have been analyzed for single muscle fibers. However, that was not practically because such single muscle fiber based analysis would

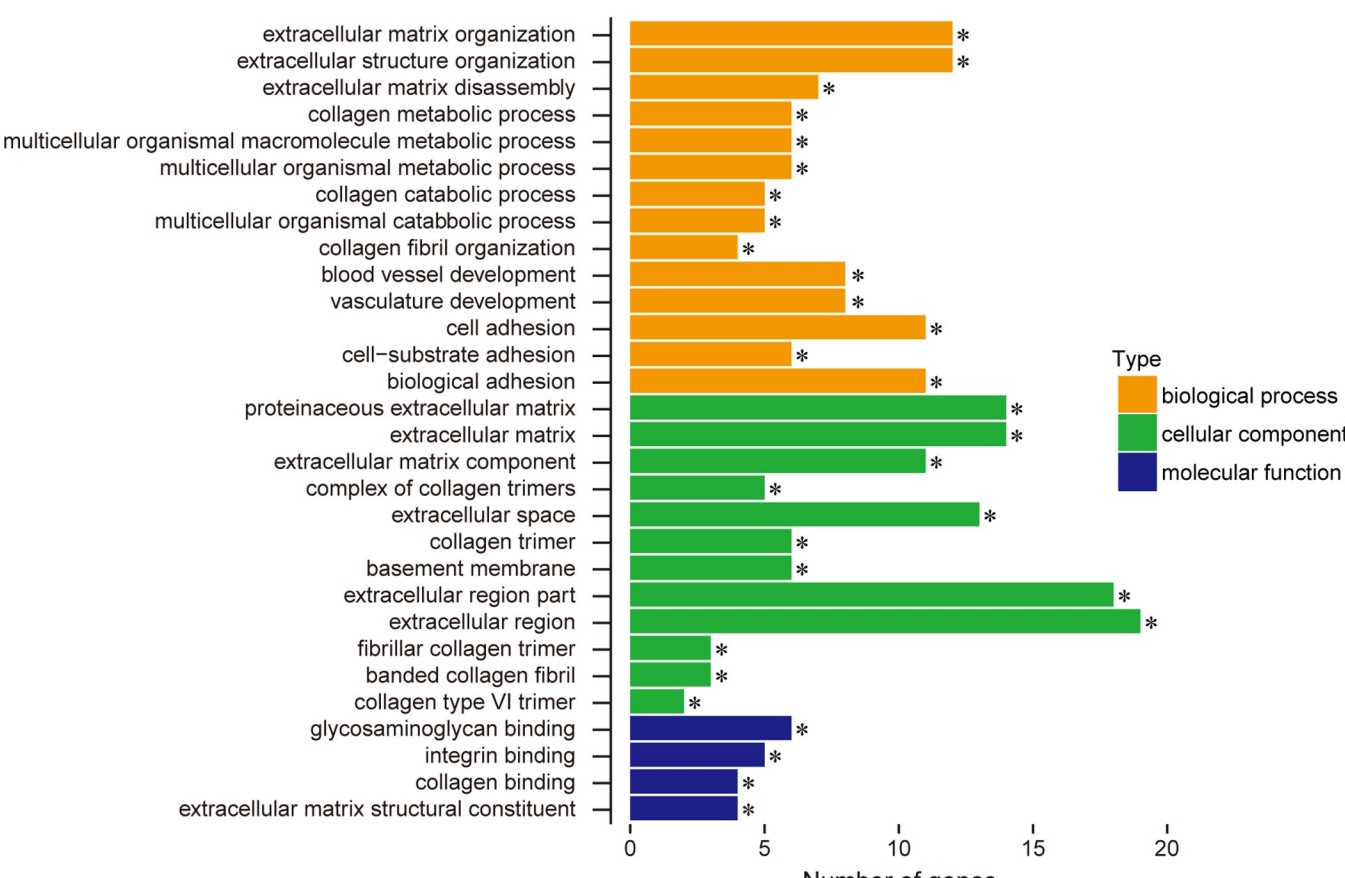

**Fig 4. Results of GO analysis in down-regulated genes.** 29 genes were targeted for analyzing their GO terms. Significant enrichment was found in 14 terms of biological process (orange), 12 terms of cellular component (green), and 4 terms of molecular function (blue). Note that the up-regulated genes frequently include the terms such as collagen, and extracellular matrix.

need more muscle tissue volume per subject. Therefore, the results of the present study are discussed as changes of the whole muscle homogenate including both slow- and fast-twitch fibers.

## Gene expression profiles

The regulation of skeletal muscle size is largely dependent on the dynamic balance between protein synthesis and degradation [15, 16]. It is also known that the mammalian target of rapamycin (mTOR) signaling pathway plays a crucial role in the skeletal muscle protein synthesis [17]. Ogasawara et al. [18] reported that the administration of mTOR kinase inhibitor, but not that of an mTOR-specific inhibitor, completely repressed the resistance exercise-induced muscle protein synthesis. We previously reported that in the same muscle samples used in the present study acute RE significantly stimulated the mTOR-induced signal cascade [13]. Furthermore, the present study showed that some factors including genes that contribute to muscle hypertrophy were up-regulated upon acute RE. For example, stress-responsive proteins coded by genes such as *Cryab*, *Dnajb*, *Hspa1a* and *Hspb1* function as molecular chaperones, are reported to be up- or down-regulated in the hypertrophic or atrophic conditions in the skeletal muscles of rodents [19–22]. It was also reported that an overexpression of the proto-oncogene *Junb* due to up-regulated upon acute RE, induced the marked hypertrophy of both

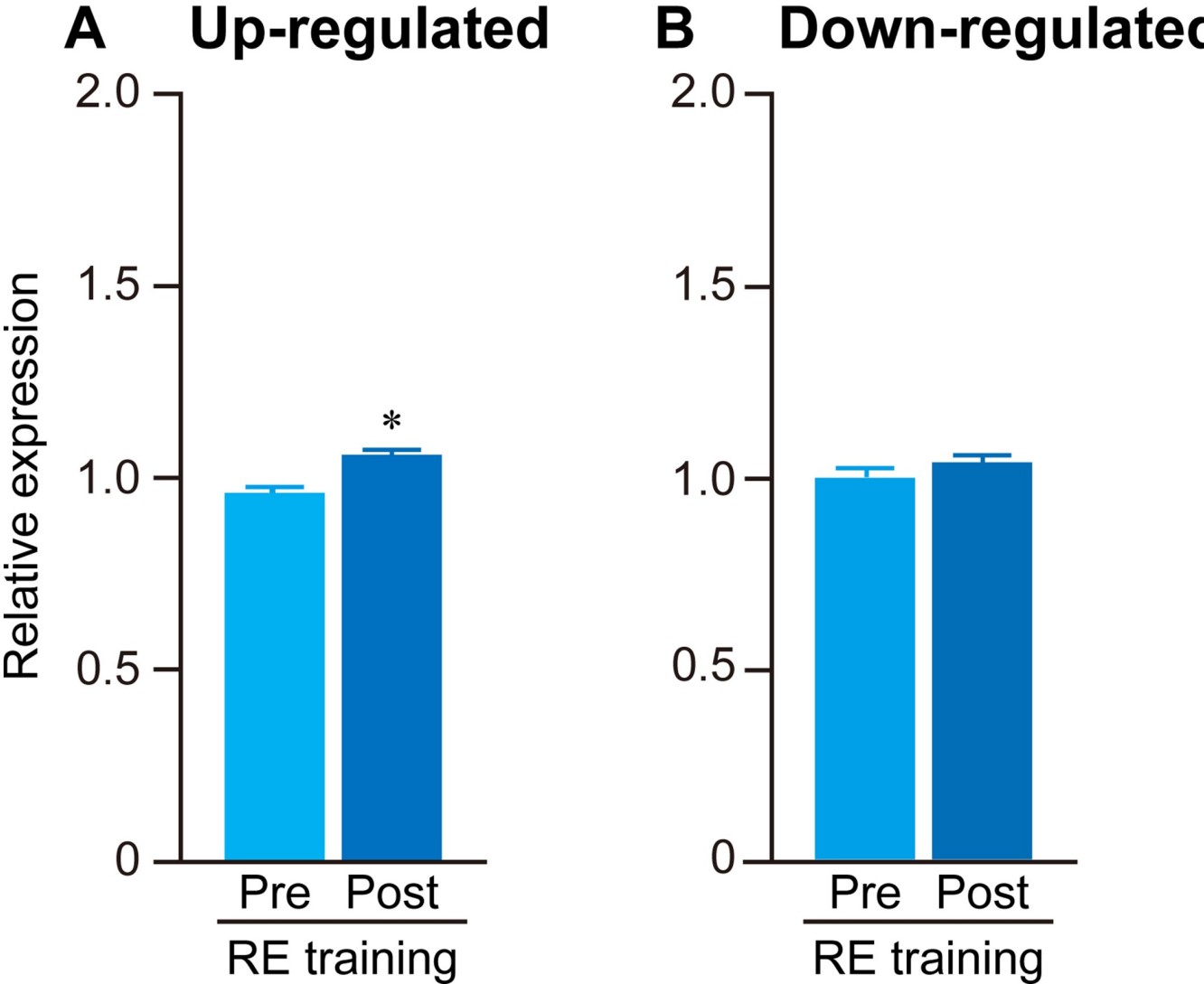

**Fig 5. Responses of gene expressions to RE training.** Expressions of genes that were up- (A) or down- (B) regulated in response to acute RE were analyzed before (Pre) and after (Post) RE training. FPKM values obtained from RNA-seq were used to calculate the mean expression levels in both groups. To normalize the differences in the distribution of data among the genes, the median value was calculated within each gene and expressed as 1. Mean ± SE. *: $p < 0.05$ vs. respective pre group, examined by paired $t$-test.

myotubes and adult mouse muscles by stimulating the overall protein synthesis and myosin expression [23]. Therefore, it was suggested that the up-regulation of these genes was closely related to the induction of anabolic pathways. *Ubc* and *Trim63*, which play a role in the proteolytic system, were also up-regulated upon acute RE. The E3 ubiquitin ligase *Trim63*, also known as muscle RING finger protein 1, was reportedly up-regulated during muscle inactivation owing to bedrest in humans [24] and tail suspension in rodents [25]. These results also indicate that RE stimulates the expression of genes which negatively regulate muscle mass.

Satellite cells, a postnatal source for growth and regeneration of skeletal muscles, differentiate into muscle fibers following proliferation and fusion and/or incorporation into the resident fibers [26]. In this experiment, the number of satellite cells and centrally nucleated fibers significantly increased after RE training, indicating that repeated RE caused muscle fiber damage [12]. Lim et al. [13] also reported that acute RE resulted in the proliferative activation of

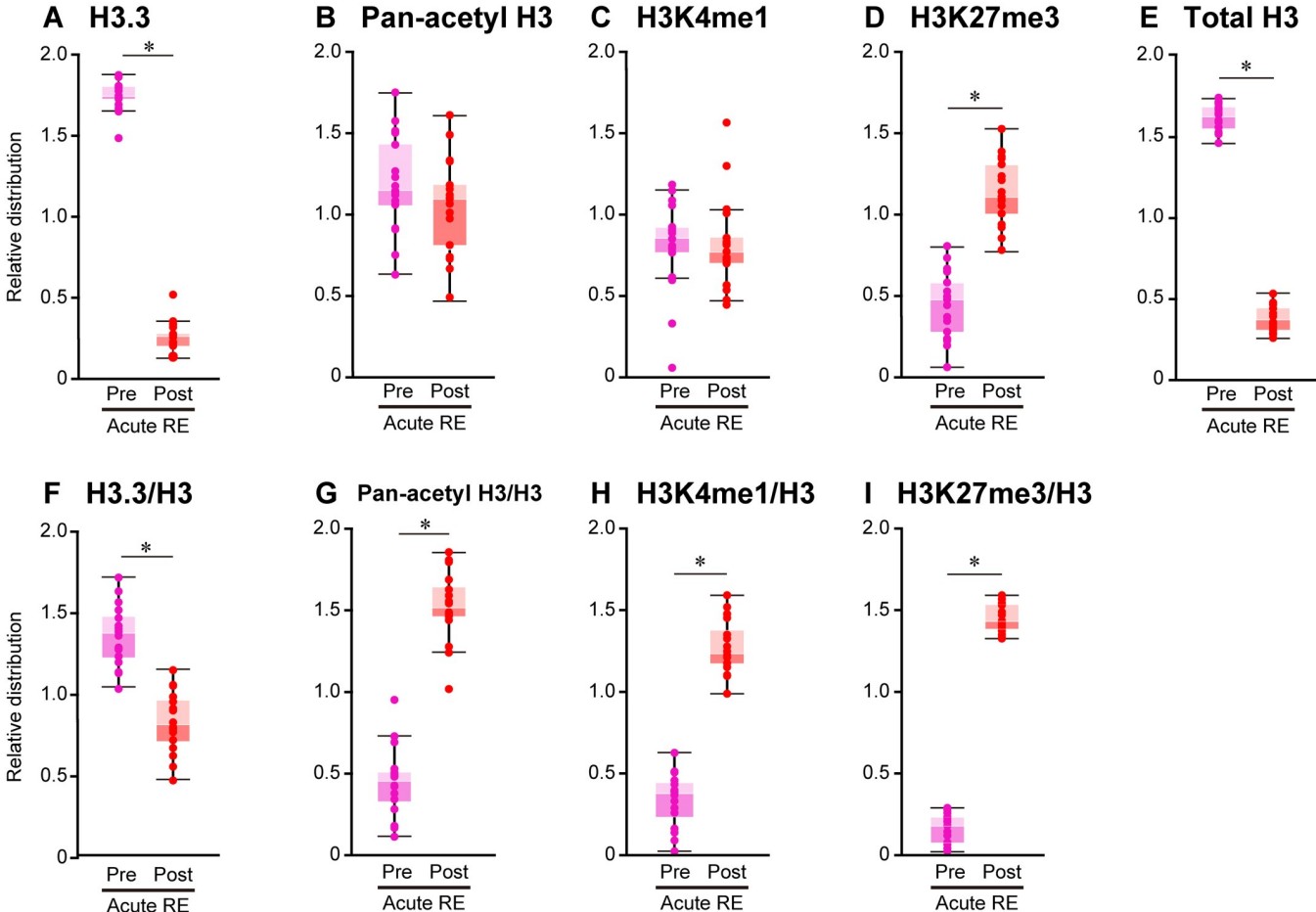

**Fig 6. Changes in histone distribution after acute RE.** Chromatin immunoprecipitation was performed with antibodies specific for H3.3 (A and F), pan-acetyl H3 (B and G), H3K4me1 (C and H), H3K27me3 (D and I), total H3 (E) followed by quantitative PCR. The distributions were analyzed at 1 kbp downstream from transcription start site of 16 target genes shown in Table 1, and expressed as a box plot. G-I: The values normalized by respective total H3 distributions. To normalize the differences in the distribution of data among the genes, the median value was calculated within each gene and expressed as 1. Mean ± SE. *: $p < 0.05$ vs. respective pre group, examined by paired $t$-test.

satellite cells. In the present study, we found that some genes with GO terms including cell death were among the genes up-regulated by acute RE. For example, *Btg2* a tumor suppressor, which arrests cells at the G1/S and the G2/M transition, increases apoptosis [27]. *Socs3* is reportedly induced by IL-6, but inhibits the IL-6-mediated JAK-STAT pathway, and suppresses inflammation [28]. These results indicate that RE led to not only muscle damage followed by inflammation but also to anti-inflammatory processes.

Interestingly, the expression of genes with GO terms which are associated with metabolism, such as *Nr4a1*, *Pdk4*, *Ppargc1a* and *Vegf*, was elevated upon acute RE in the present study. It was reported that these genes were also up-regulated in the human skeletal muscle upon endurance exercise, and are closely related to mitochondrial biogenesis [29–32]. Porter et al. [33] reported that 12 weeks of RE training enhanced mitochondrial coupled respiration in human skeletal muscle, a finding that could be explained by an increased expression of complex I proteins. These results indicate that RE also stimulates pathways enhancing endurance capacity in skeletal muscle, although it is unknown whether this signaling is necessary to induce skeletal muscle hypertrophy, or is an additional response to RE.

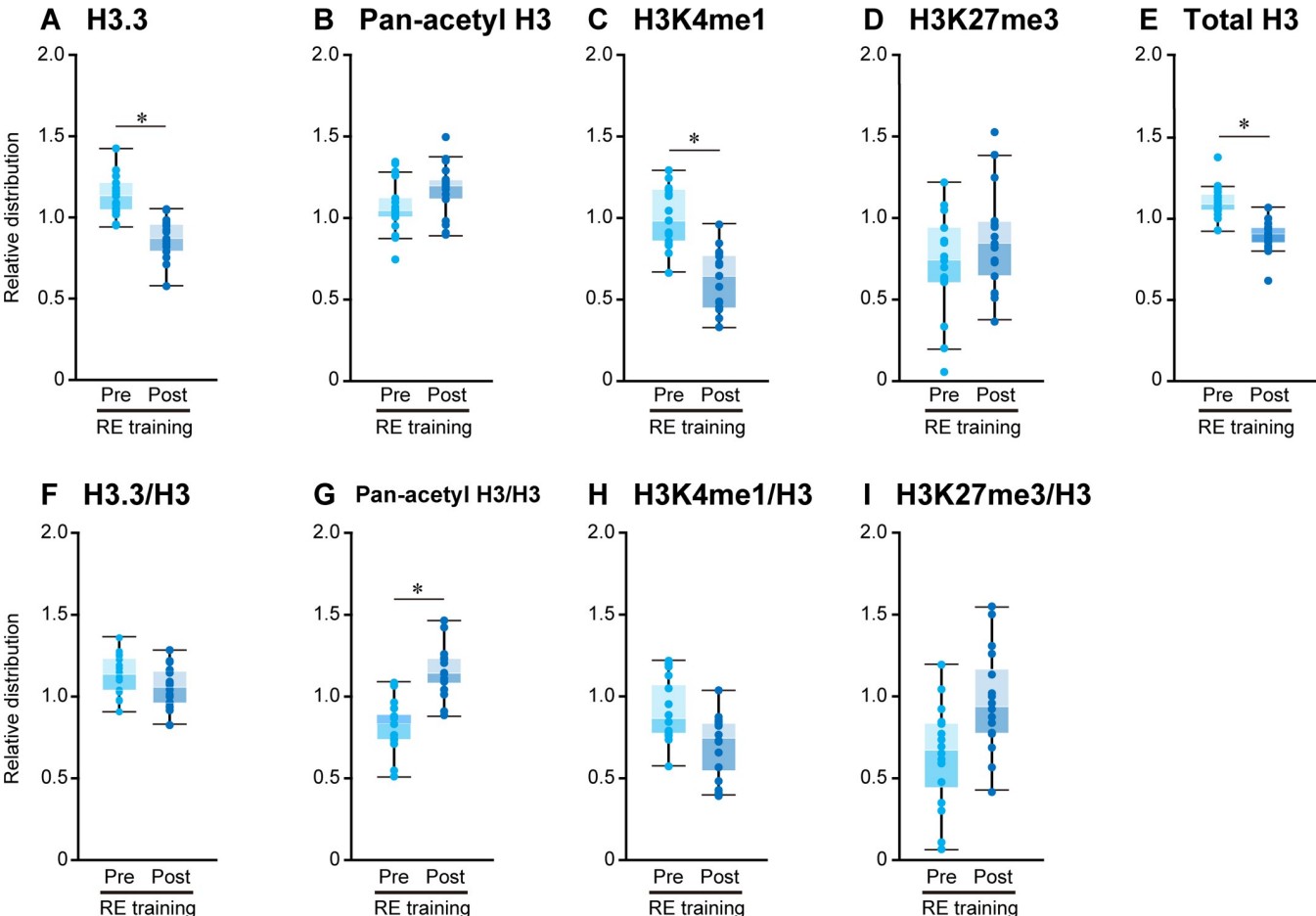

**Fig 7. Changes in histone distribution after RE training.** Chromatin immunoprecipitation was performed with antibodies specific for H3.3 (A and F), pan-acetyl H3 (B and G), H3K4me1 (C and H), H3K27me3 (D and I), total H3 (E) followed by quantitative PCR. See Fig 6 for the details. Mean ± SE. $^*$: $p < 0.05$ vs. respective pre group, examined by paired $t$-test.

It has been reported that chronic RE training induces skeletal muscle hypertrophy in both human [3, 8, 12] and rodent [34] models. The present study successfully induced hypertrophy of muscle fibers after RE training, based on the histochemical analysis results of the muscle samples used for the present study [12]. The present study also showed that transcription of the genes up-regulated upon acute RE was activated after RE training. This shows that repeated RE enhanced the responses of genes to physiological stimuli and is linked to the hypertrophy of muscle fibers.

## Epigenetic regulations

Results of the present study showed a drastic loss of total H3 levels at the loci transcriptionally activated after acute RE (Fig 6E). We speculate that nucleosomes were dissembled in association with the up-regulation of gene transcription upon RE. The distribution of histone H3.3 variant also decreased at the loci after acute RE (Fig 6A and 6F). These results suggest that dissembling nucleosomes preceded transcriptional activation, which was stimulated by the dissociation of H3.3. We previously reported that endurance training decreased nucleosome formation and promoted incorporation of histone H3.3 variant into the nucleosomes in the plantaris muscle of adult rats, indicating that exercise training stimulates the turnover of

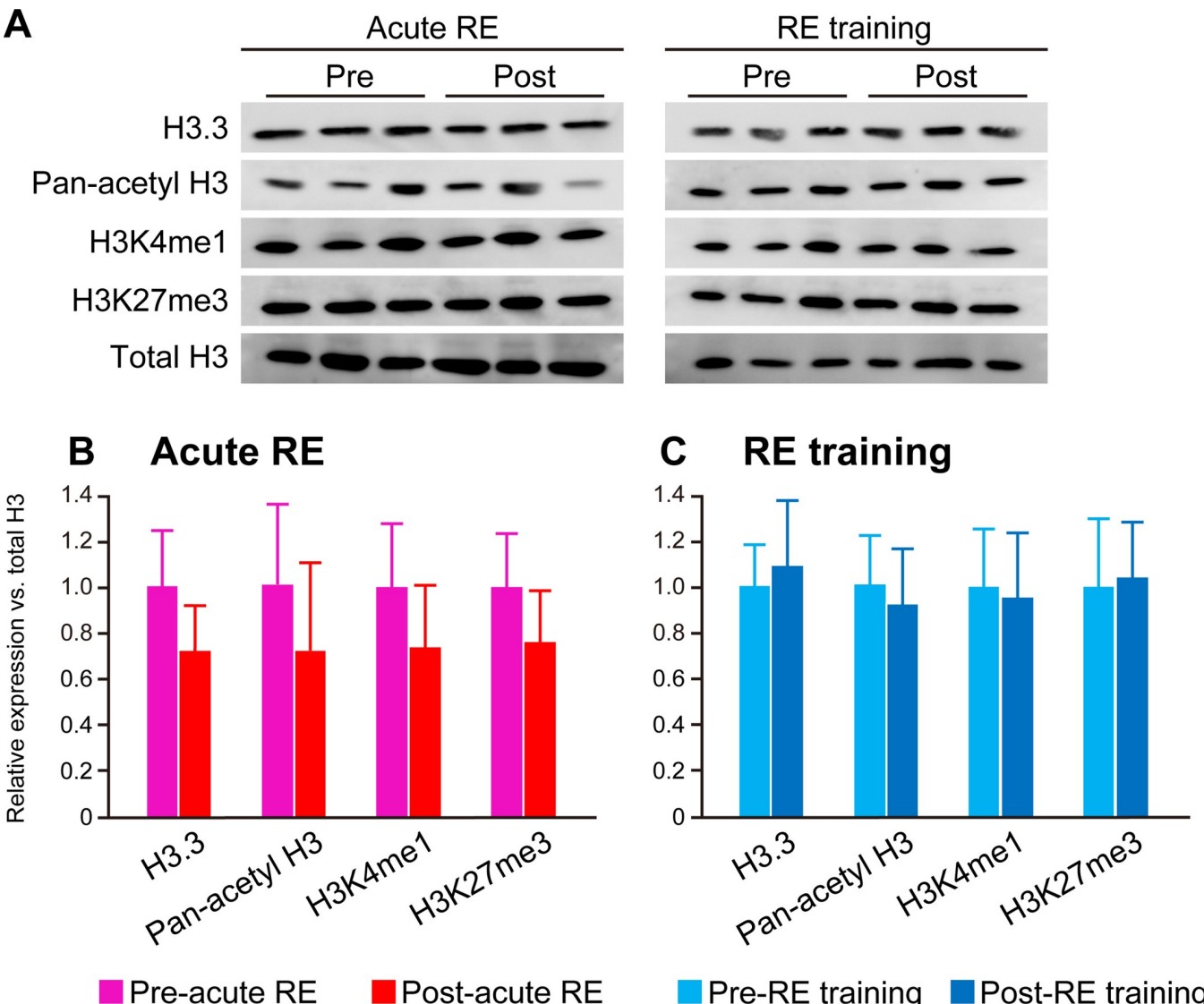

**Fig 8. Effects of acute RE and RE training on protein expressions.** A: Typical band images of histones obtained in western blotting. Same samples used in Figs 6 and 7 were analyzed. B and C: Quantitative analysis of band intensities in Experiment 1 (acute RE) (B) and 2 (RE training) (C). The values were normalized into the mean level of respective pre group was 1.

histones [7, 35]. However, results of the present study suggested that RE training did not cause the turnover of histones, although the number of nucleosomes was lower after RE training.

Acetylation of histone residues is known as a modification in the transcriptional activation, and is associated with euchromatin. Previous studies [36, 37] have demonstrated that hyperacetylation of histones was induced at the loci of genes transcriptionally activated after a single bout of swimming exercise. Smith et al. [36] reported that activation of the calcium/calmodulin-dependent protein kinase was indispensable for the hyperacetylating and binding of myocyte enhancer factor 2A at the glucose transporter 4 promoter, which preceded up-regulation of the glucose transporter 4 gene expression in response to exercise. Results of the present study also showed an increase in H3 acetylation after acute RE (Fig 6B and 6G), suggesting that histone acetylation occurs at the loci transcriptionally activated upon exercise regardless

of the type of exercise. Furthermore, Ohsawa et al. [35] demonstrated that exercise training promotes the turnover of histones at the transcriptionally activated loci of rat skeletal muscle and inhibited the accumulation of acetylated histones after long-term training; although skeletal muscle of rats displayed lower turnover of histones and acetylated histones were accumulated in this tissue. This result also supports our suggestion that the limitation in histone turnover actually promoted the accumulation of acetylated histones during repeated RE stimulation. The accumulation of acetylated histones might reduce nucleosome formation, maintain the loci at transcriptionally active status, and enhance gene expressions after RE training. We also observed that the increase in histone acetylation was not resulted from the changes in the net acetylation rate, which is in accordance with the expression levels of histone modifications were not affected by RE as seen in western blotting results.

H3K4me1 and H3K27me3 are positively and negatively correlated with gene transcription levels in a wide range of cell types, respectively [38]. Blum et al. [39] reported that both H3K4me1 and acetylation at H3K27 are highly conserved at the myogenic factor MyoD-binding sites in the myoblasts and myotubes, leading to the muscle-specific gene expression. Acute RE promoted both H3K4me1 and H3K27me3 at the transcriptionally activated loci, suggesting that these loci were bivalently modified during transcriptional activation. However, the distributions of H3K4me1 and H3K27me3 at these loci were not significantly changed after RE training (Fig 7H and 7I). The results again suggest that the increased distribution of acetylated H3 is closely related to the enhanced expressions in a subset of genes after RE training.

Recently, it has been reported that DNA methylation is also affected by RE in human skeletal muscles [8, 9]. Seaborne et al. [8] reported that the frequency of genome-wide hypomethylation at the CpG sites in human skeletal muscle was increased after chronic RE training (9,153 sites), maintained during the detraining period (8,891 sites), and further enhanced in response to reloading (18,816 sites). Turner et al. [9] further reported that the expression of 592 of 5,262 genes that were hypomethylated at their CpG sites after chronic RE training was up-regulated. These results indicate that RE training altered the status of genes activation, in agreement with our present results which show an enhancement of the hallmarks of active transcription in histone distributions after RE training.

The up-regulation of protein synthesis in rat skeletal muscle in response to RE gradually decreased during the period of RE training [40, 41]. These studies also reported that acute responses of mTOR signaling, such as the phosphorylation of p70S6K and ribosomal protein S6, were reduced if the RE stimulus was repeatedly provided. These results indicate a reduced responsiveness of the skeletal muscle after chronic RE training, whereas the results of previous studies [8, 9] and our present study consistently show that there is an enhancement of the transcriptionally active hallmarks in the epigenome of skeletal muscle after RE training. When RE training leads to an increased transcriptional activation of genes; genes that negatively regulate the skeletal muscle mass are also considered to be transcriptionally activated in the late period of training upon RE stimulation. Further studies are necessary to understand the relationship between epigenome and responsiveness of genes, and the associated changes in the epigenome after exercise training.

## Supporting information

**S1 File.**
(ZIP)

**S2 File.**
(XLSX)

**S3 File.**
(XLSX)

**S1 Raw images.**
(PDF)

## Author Contributions

**Conceptualization:** Changhyun Lim, Fuminori Kawano, Hyo Jeong Kim, Chang Keun Kim.

**Data curation:** Changhyun Lim, Junya Shimizu, Fuminori Kawano.

**Formal analysis:** Changhyun Lim, Junya Shimizu, Fuminori Kawano.

**Funding acquisition:** Fuminori Kawano, Hyo Jeong Kim, Chang Keun Kim.

**Investigation:** Changhyun Lim, Junya Shimizu, Fuminori Kawano.

**Methodology:** Changhyun Lim, Fuminori Kawano, Hyo Jeong Kim, Chang Keun Kim.

**Project administration:** Hyo Jeong Kim, Chang Keun Kim.

**Supervision:** Hyo Jeong Kim, Chang Keun Kim.

**Validation:** Chang Keun Kim.

**Visualization:** Junya Shimizu, Fuminori Kawano.

**Writing – original draft:** Junya Shimizu, Fuminori Kawano.

**Writing – review & editing:** Changhyun Lim, Fuminori Kawano, Chang Keun Kim.

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
