## [Decision Letter · Decision Letter 0]

12 Dec 2019

PONE-D-19-25531

Adaptive responses of histone modifications to resistance exercise in human skeletal muscle

PLOS ONE

Dear Dr Kawano,

Thank you for submitting your manuscript to PLOS ONE. After careful consideration, we feel that it has merit but does not fully meet PLOS ONE’s publication criteria as it currently stands. Therefore, we invite you to submit a revised version of the manuscript that addresses the points raised during the review process.

Specifically, the experts have suggested ways to improve the abstract , clarify the Methods and providing more detail for the statistics and clarify several places in the discussion. The Reviewers have done a nice job of providing the concerns and ways to address them. Finally, the figures should be redone at a high resolution to make them clear.

We would appreciate receiving your revised manuscript by Jan 26 2020 11:59PM. To enhance the reproducibility of your results, we recommend that if applicable you deposit your laboratory protocols in protocols.io, where a protocol can be assigned its own identifier (DOI) such that it can be cited independently in the future. For instructions see: http://journals.plos.org/plosone/s/submission-guidelines#loc-laboratory-protocols

We look forward to receiving your revised manuscript.

Kind regards,

Stephen E Alway, Ph.D.

Academic Editor

PLOS ONE

Journal Requirements:

1. We note that you have indicated that data from this study are available upon request. PLOS only allows data to be available upon request if there are legal or ethical restrictions on sharing data publicly. For more information on unacceptable data access restrictions, please see http://journals.plos.org/plosone/s/data-availability#loc-unacceptable-data-access-restrictions.

4. Please include your tables as part of your main manuscript and remove the individual files. Please note that supplementary tables (should remain/ be uploaded) as separate "supporting information" files

Reviewers' comments:

Reviewer's Responses to Questions

**Comments to the Author**

1. Is the manuscript technically sound, and do the data support the conclusions?

Reviewer #1: Yes

Reviewer #2: Yes

2. Has the statistical analysis been performed appropriately and rigorously? 

Reviewer #1: No

Reviewer #2: Yes

3. Have the authors made all data underlying the findings in their manuscript fully available?

Reviewer #1: Yes

Reviewer #2: Yes

4. Is the manuscript presented in an intelligible fashion and written in standard English?

Reviewer #1: Yes

Reviewer #2: Yes

5. Review Comments to the Author

Reviewer #1: Lim and colleagues investigated what genes are altered in human skeletal muscle in response to an acute bout of resistance exercise or resistance exercise training using RNA-seq. qPCR and Western blot was used to validate the response in a subset of genes and/or proteins. Many genes showed similar expression patterns following acute resistance exercise and resistance exercise training. The authors reported intriguing changes in histones that are related to the active genes.

The technical aspects of this manuscript are quite good. The data is physiologically relevant as coming from human muscle biopsies. The manuscript is written in a manner that is easy to read and understand. There are some limitations to the study, but a good follow-up approach maximizing samples from another published work.

The following are some recommendations to strengthen the manuscript for publication:

1) Abstract: The abstract reads a little ambiguous. Can the authors provide percent changes and p-values to the abstract? The conclusion statement is a bit strong. The results suggest histone acetylation may play a role in the active status of genes, but remember, the current data is correlative regarding that matter, not cause and effect.

2) Methods: the methodology relating to the biopsy samples needs to be detailed. What time after the last bout of exercise was the biopsy taken? What are the differences between the 80FAIL, 30WM, and 30FAIL? What was the statistical measure used to determine these groups could be grouped together, from which 9 biopsies in total were analyzed for this current study? While 9 biopsies were taken at random, post-hoc analysis can now be used to determine any variability in the samples related to the outcome measurements of the subjects from the first study.

3) Statistics: the statistics for RNA-Seq, ChIP, typically are more complex than those reported in the statistical analysis section. Please provide enough detail so that the RNA-seq data could be analyzed to a similar conclusion in different hands. Is the paired t-test used because of the within-subjects analysis, i.e., biopsies from same individuals at different times? Please add more detail for clarity

4) Discussion: please clarify when data is being discussed from the previous publication related to these biopsy samples. I think the discussion requires a Limitations paragraph that addresses the sample decisions made by the authors. For example, only 9 biopsies from 30 participants, and only 3 biopsies analyzed for ChIP and Western Blot. This reduces the Power of the study and should be acknowledged. And again, I think the conclusion needs to be less forceful. Even if all the samples were analyzed, the data is still a relationship between histone acetylation, exercise, and gene expression, not cause and effect.

5) Figures: all the figures downloaded a little blurry. Might have been on my end, but please check the resolution.

Reviewer #2: General Comments:

This study investigated skeletal muscle histone modifications in two groups of men: One group Pre- vs Post- acute RE, and another group Pre- vs Post- RE training.

Understanding the muscle epigenetic responses to RE in humans is an emerging topic, and the authors are commended on this work. However, a more elegant study design could have been conducted in this case. While the data presented here is useful, a more controlled approach would have been to: Do “pre- vs post- acute RE” biopsies, then do an RE training intervention on all men, then do “pre- vs post- acute RE” biopsies after RE training as well (4 time points in the same group). In the current article, authors should make it more clear that this is really two separate groups being studied, and thus, two separate experiments. Simple changes in the manuscript wording could be implemented to make this distinction, such as adding numbers in the abstract to show there are two separate experiments: “The present study was carried out to investigate the effects of 1) acute RE and 2) RE training on gene expression profiles and histone modifications in human skeletal muscle.”

That said, I still feel that this work is well written, and data are well presented. The article will be a good addition to the human RE epigenetics literature.

Specific comments/suggestions:

Please add page numbers to make review more efficient.

Abstract:

Please indicate the number of subjects, age, height, and body mass of each group of men. Example: “Healthy male adults were assigned to acute RE (n= , age= y, BIM= kg/m2) or RE training (n= , age= y, BIM= kg/m2) groups.”

Please add more informative data to the results sections of the abstract. Please indicate the fold-increase/decrease (or percent change) of significant variables to give the readers more information up front. Example: “Up-regulation of acetylated histone 3 (H3) (+235%) and H3 …”.

Introduction:

Paragraph 1, Line 3: Amend sentence to: “…muscle fiber hypertrophy induced upon by resistance exercise (RE) training.”

Paragraph 2: It is described here that different fiber types show different histone modification patterns. Fiber type distributions differ significantly among humans (even in the same muscle), and fiber type was not reported in the current article. Homogenizing samples with different fiber type proportions could have confounding effects on the data. In future studies, fiber type should be reported between pre vs post RE training groups, or single fibers should be isolated and pooled to get a more accurate representation of what is happening at the muscle cell level (see this paper that did fiber type specific analysis of DNA methylation in human muscle: (Begue, et al. 2017: https://www.physiology.org/doi/full/10.1152/japplphysiol.00867.2016 ). The limitation of analyzing homogenized samples should be mentioned in the Methods or discussion.

In the last paragraph of the introduction, there is a disconnect between going from animal studies, to human studies. I feel that a sentence or two need to be added stressing that very few studies have been conducted investigating epigenetics in human muscle after acute RE and/or RE training.

Example of last paragraph highlighting the little work done in humans: “We further reported that endurance exercise training decreased the responsiveness of genes to muscular unloading in association with enhanced incorporation of histone variant H3.3 into nucleosomes in the plantaris muscle of rats. This indicated that endurance exercise training stimulated turnover of histones [6], but it is currently unclear how RE affects the distribution of histones and their modification patterns in animal models. Furthermore, even less research has been conducted on the epigenetic response to RE in human skeletal muscle (Cite 29, 36, article below). Therefore, the present study built upon the animal literature, and aimed to investigate the response of histones in human muscle before and after 1) acute RE and 2) chronic RE.”

Cite these articles on epigenetics after RE in humans:

Bagley et al. (2019) https://journals.lww.com/nsca-jscr/Abstract/publishahead/Epigenetic_Responses_to_Acute_Resistance_Exercise.94813.aspx

Romero et a. (2018) https://www.ncbi.nlm.nih.gov/pubmed/29351416

Statistical Analysis:

Please indicate the static at software package used to analyze these data.

Discussion:

Paragraph 1: Add a period after this sentence: “however, are reportedly up- or downregulated

in hypertrophic or atrophic conditions in skeletal muscles of rodents [14-17].”

Figures:

Write “RE Training” throughout all figures (to be similar to “Acute AE”).

6. PLOS authors have the option to publish the peer review history of their article (what does this mean?). If published, this will include your full peer review and any attached files.

Reviewer #1: Yes: Jarrod A Call, PhD

Reviewer #2: Yes: James R. Bagley

---

## [Author Response · Author response to Decision Letter 0]

24 Dec 2019

Responses to the Reviewer #1

1) Abstract: The abstract reads a little ambiguous. Can the authors provide percent changes and p-values to the abstract? The conclusion statement is a bit strong. The results suggest histone acetylation may play a role in the active status of genes, but remember, the current data is correlative regarding that matter, not cause and effect.

Responses:

Abstract was revised as including the percent changes and p values. Sentences on the bottom of abstract were also re-phrased stating conclusion directly suggested by the current results as “These results indicated that a single bout of RE drastically alter both gene expressions and histone modifications in human skeletal muscle. It was also suggested that enhanced histone acetylation closely related to up-regulation of gene expressions after RE training.” in line 17-19 of page 2.

2) Methods: the methodology relating to the biopsy samples needs to be detailed. What time after the last bout of exercise was the biopsy taken? What are the differences between the 80FAIL, 30WM, and 30FAIL? What was the statistical measure used to determine these groups could be grouped together, from which 9 biopsies in total were analyzed for this current study? While 9 biopsies were taken at random, post-hoc analysis can now be used to determine any variability in the samples related to the outcome measurements of the subjects from the first study.

Responses:

More precise statements were added to find the differences between 80FAIL, 30WM, and 30FAIL groups as “The 80FAIL group exercised at 80% of 1RM until they could not keep producing muscular contractions in every set, the 30WM group performed the total workload matching that of the 80FAIL, and the 30FAIL group exercised at 30% of 1RM until failure. For example, total work volume performed in leg press was 2,358-3,030, 2,495-3,060, and 2,995-3,588 kg/set in 80FAIL, 30WM, and 30FAIL, respectively (see Ref. 12 for more details).” in line 5-10 of page 6.

Experiment 1 (acute RE) and 2 (RE training) used different subjects. In Experiment 1, all biopsies (n=9) were combined to use for gene expression analysis, and 3 of 9 biopsies were chosen for histone modification analysis. In Experiment 2, 9 biopsies were randomly selected from 21 subjects for the gene expression analysis, and 3 of 9 biopsies were chosen for histone modification analysis. The effect of exercise method and time on the muscle fiber size was examined in Ref. 12, which showed that the two variables did not cause any interactive effect on the muscle fiber size (F=2.448, p=0.115). Therefore, the main effect test was done, which found difference by time (F=20.831, p<0.001), so that the present study selected the samples for analysis randomly from 3 groups. The sentences were revised as “Nine biopsy samples were randomly collected from 3 groups (21 subjects) as samples from individuals subjected to RE training, and were used to analyze gene expressions in the present study, because the main effect test by two-way ANOVA tested in all groups found difference in muscle fiber size by time (pre vs. post) (see Ref. 12 for precise results of each group). Further, 3 of 9 biopsies were selected to analyze histone modifications. Throughout the analysis of biopsies obtained from Experiment 2, biopsies of pre- and post-RE training were selected from same subjects.” in line 13-19 of page 6.

3) Statistics: the statistics for RNA-Seq, ChIP, typically are more complex than those reported in the statistical analysis section. Please provide enough detail so that the RNA-seq data could be analyzed to a similar conclusion in different hands. Is the paired t-test used because of the within-subjects analysis, i.e., biopsies from same individuals at different times? Please add more detail for clarity

Responses:

Statistics section was revised as “Statistical analysis was performed using BellCurve for Excel (Social Survey Research Information Co., Ltd.). For the data analysis of RNA-seq, all FPKM values were compared in same genes between all experimental groups, and calculate a correlation coefficient (R2) (Fig. 1). FPKM values were normalized using the median within each gene, and averaged in up- or down-regulated genes (Fig. 2). For ChIP data, a boxplot was used to display the distribution of the data obtained from each gene (Figs. 5 and 6). Values plotted in the figures 2, 5, 6, and 7 were compared to determine the significant differences only between pre- and post-acute RE or RE training. Because the data of pre- and post-groups were obtained from same subjects in both Experiment 1 and 2, significant differences were examined using a paired t-test. Differences were considered significant at p<0.05.” in line 21 of page 9 to line 3 of page 10.

4) Discussion: please clarify when data is being discussed from the previous publication related to these biopsy samples. I think the discussion requires a Limitations paragraph that addresses the sample decisions made by the authors. For example, only 9 biopsies from 30 participants, and only 3 biopsies analyzed for ChIP and Western Blot. This reduces the Power of the study and should be acknowledged. And again, I think the conclusion needs to be less forceful. Even if all the samples were analyzed, the data is still a relationship between histone acetylation, exercise, and gene expression, not cause and effect.

Responses:

Thank you for the suggestion. New sub-section “Limitation of analysis” was added in discussion, and stated as “The present study shared human biopsy samples with the original research group, whose data were published in Ref. 12 and 13. Since scientific significance would be maximized if histone modifications were measured using the samples that was enough analyzed in histochemical characteristics, and protein expressions, the present study was motivated to analyze the remaining biopsy samples. However, because of limited volume of samples remaining, we could not analyze all samples individually. Therefore, 9 biopsies were randomly chosen in Experiment 2, furthermore, 3 of 9 biopsies were selected for ChIP and western blot analysis in both experiments. But, the biopsies of pre- and post-groups in both Experiment 1 and 2 were selected from same subjects.

 Epigenetic regulation differs between muscle fiber types [6]. For an essential manner analyzing data, in the present study histone modifications should have analyzed in single muscle fibers. However, that was technically difficult because the analysis needed much tissue volume. Therefore, the results of the present study discuss as changes of whole muscle homogenate including both slow- and fast-twitch fibers.” in line 2-14 of page 12.

 Discussion was revised to that the data were discussed a relationship between histone acetylation, exercise, and gene expression. Further, discussion was revised to avoid a direct relationship between the results of acute RE and RE training, because these experiments targeted different subjects.

5) Figures: all the figures downloaded a little blurry. Might have been on my end, but please check the resolution.

Responses:

 I have confirmed that current version of figures was rendered by 600 dpi resolution.

 

Responses to the Reviewer #2

Responses to the general comments:

 Thank you for positive comments. As pointed by the reviewer, manuscript was revised as that acute RE and RE training were completely separated into two different experiments. Sentence in introduction was revised as “Therefore, the present study built upon the animal literature, and aimed to investigate the response of gene expression profiles and histone modifications in human muscle before and after 1) acute RE and 2) chronic RE training.” in line 2-5 of page 4. Experimental designs were shown in methods as “

Experiment 1

 Subjects: Nine male weight lifters (age=20.5±4.3yr, BMI=28.0±6.8kg/m2, mean ± SD) participated in Experiment 1 (acute RE). All subjects were national caliber weightlifters of H City team including a London Olympic medalist. They had been training at least 7 years and had a similar living condition such as diet, nutrition and living in the same dormitory.

 Exercise protocol: A week prior to the test, subjects completed a 10 repeated maximum (RM) squat and bench press test to ensure appropriate exercise intensity (172.2±38kg, 82.6±16.4kg, 1RM of squat and bench press, mean ± SD, respectively). Intensity of exercise was then set at 60% of their 1RM weight and subjects competed 3 sets of 6 repetitions during each session of exercise. RE consisted of squat, single leg lunge, and deadlift which was repeated twice by all subjects.

 Collection of muscle biopsy samples: Muscle biopsy samples were obtained using local anesthesia (1% lidocaine) administrated into the mid belly of the vastus lateralis muscle immediately before (pre-acute RE), and 3 hours (post-acute RE) after exercise. Muscle biopsy samples were frozen in liquid nitrogen and stored at −80°C until analysis. Nine biopsy samples were combined to analyze gene expressions and histone modifications in Experiment 1. Further, 3 of 9 biopsies were selected to analyze histone modifications. Throughout the analysis of biopsies obtained from Experiment 1, biopsies of pre- and post-acute RE were selected from same subjects.

Experiment 2

 Subjects: Effects of RE training were examined in 21 males (age=23.7±2.5yr, BMI=24.2±2.7kg/m2) who had no special medical disorders in musculoskeletal, cardiovascular, and respiratory systems and had not done any regular resistance exercise in the last 2 years. The subjects were separated into 3 groups; 80FAIL (n=7, age=24.5±1.8yr, BMI=25.9±3.9kg/m2), 30WM (n=7, age=23.1±2.0yr, BMI=25.0±3.1kg/m2), and 30FAIL (n=7, age=23.0±1.2yr, BMI=24.4±1.3kg/m2).

 Exercise protocol: All subjects performed 3 repeated sets of leg press, leg extension, and leg curl three times per week for 10 weeks in Experiment 2. Considering the risk of injury to subjects who had not performed resistance exercise before, 1RM was determined by using the indirect measurement method suggested in a previous report [14]. The 80FAIL group exercised at 80% of 1RM until they could not keep producing muscular contractions in every set, the 30WM group performed the total workload matching that of the 80FAIL, and the 30FAIL group exercised at 30% of 1RM until failure. For example, total work volume performed in leg press was 2,358-3,030, 2,495-3,060, and 2,995-3,588 kg/set in 80FAIL, 30WM, and 30FAIL, respectively (see Ref. 12 for more details).

Collection of muscle biopsy samples: Muscle biopsy was sampled from the mid belly of the vastus lateralis muscle by aforementioned procedures prior to training (pre-RE training), and 72 hours after the final session of RE (post-RE training). Nine biopsy samples were randomly collected from 3 groups (21 subjects) as samples from individuals subjected to RE training, and were used to analyze gene expressions in the present study, because the main effect test by two-way ANOVA tested in all groups found difference in muscle fiber size by time (pre vs. post) (see Ref. 12 for precise results of each group). Further, 3 of 9 biopsies were selected to analyze histone modifications. Throughout the analysis of biopsies obtained from Experiment 2, biopsies of pre- and post-RE training were selected from same subjects.” in line 5 of page 5 to line 19 of page 6.

Comments:

Please add page numbers to make review more efficient.

Responses:

 We are sorry to miss page numbers. The revised version of manuscript was added page numbers.

Comments:

Please indicate the number of subjects, age, height, and body mass of each group of men. Example: “Healthy male adults were assigned to acute RE (n= , age= y, BIM= kg/m2) or RE training (n= , age= y, BIM= kg/m2) groups.”

Responses:

 Physical information of subjects were added in abstract as “Healthy male adults were assigned to acute RE (n=9, age=20.5±4.3yr, BMI=28.0±6.8kg/m2) or RE training (n=21, age=23.7±2.5yr, BMI=24.2±2.7kg/m2) groups.” in line 5-7 of page 2.

Comments:

Please add more informative data to the results sections of the abstract. Please indicate the fold-increase/decrease (or percent change) of significant variables to give the readers more information up front. Example: “Up-regulation of acetylated histone 3 (H3) (+235%) and H3 …”.

Responses:

 Abstract was revised to include more informative data as “RNA sequencing analysis revealed that 153 genes with GO terms including muscle development, stress response, metabolism, cell death, and transcription factor were significantly up-regulated (+291% vs. pre-acute RE) upon acute RE. Expressions of these genes were also greater (+9.6% vs. pre-RE training, p<0.05) after RE trained subjects. Significant up-regulation of acetylated histone 3 (H3) (+235%) and H3 mono-methylated at lysine 4 (+290%) and tri-methylated at lysine 27 (+849%), and down-regulation of H3.3 variant (−39%) distributions relative to total H3 were observed at transcriptionally activated loci after acute RE compared to pre-acute RE levels. Interestingly, the distribution of acetylated H3 was found to be up-regulated as compared to the level of total H3 after RE training (+40%, p<0.05).” in line 8-17 of page 2.

Comments:

Paragraph 1, Line 3: Amend sentence to: “…muscle fiber hypertrophy induced upon by resistance exercise (RE) training.”

Responses:

 The sentence was revised to “Bammam et al. [1] reported that subjects were classified into extreme responders, modest responders, and non-responders depending on the magnitude of muscle fiber hypertrophy induced upon by resistance exercise (RE) training.” in line 2-5 of page 3.

Comments:

Paragraph 2: It is described here that different fiber types show different histone modification patterns. Fiber type distributions differ significantly among humans (even in the same muscle), and fiber type was not reported in the current article. Homogenizing samples with different fiber type proportions could have confounding effects on the data. In future studies, fiber type should be reported between pre vs post RE training groups, or single fibers should be isolated and pooled to get a more accurate representation of what is happening at the muscle cell level (see this paper that did fiber type specific analysis of DNA methylation in human muscle: (Begue et al. 2017). The limitation of analyzing homogenized samples should be mentioned in the Methods or discussion.

Responses:

 Sentences were added in second paragraph of introduction as “Begue et al. [6] demonstrated fiber type-specific DNA methylation in human skeletal muscle, showing CpG sites of genes selectively expressed in type 1 or IIa myosin heavy chain fibers were hypomethylated. These data suggested that epigenetic regulation based on muscle fiber type characteristics affected the responsiveness of genes to exercise.” in line 20-23 of page 3. 

New sub-section “Limitation of analysis” was also added in discussion, and stated as “The present study shared human biopsy samples with the original research group, whose data were published in Ref. 12 and 13. Since scientific significance would be maximized if histone modifications were measured using the samples that was enough analyzed in histochemical characteristics, and protein expressions, the present study was motivated to analyze the remaining biopsy samples. However, because of limited volume of samples remaining, we could not analyze all samples individually. Therefore, 9 biopsies were randomly chosen in Experiment 2, furthermore, 3 of 9 biopsies were selected for ChIP and western blot analysis in both experiments. But, the biopsies of pre- and post-groups in both Experiment 1 and 2 were selected from same subjects.

 Epigenetic regulation differs between muscle fiber types [6]. For an essential manner analyzing data, in the present study histone modifications should have analyzed in single muscle fibers. However, that was technically difficult because the analysis needed much tissue volume. Therefore, the results of the present study discuss as changes of whole muscle homogenate including both slow- and fast-twitch fibers.” in line 2-14 of page 12.

Comments:

In the last paragraph of the introduction, there is a disconnect between going from animal studies, to human studies. I feel that a sentence or two need to be added stressing that very few studies have been conducted investigating epigenetics in human muscle after acute RE and/or RE training.

Responses:

 Thank you for suggesting the example of sentences. The last paragraph of introduction was revised as “We further reported that endurance exercise training decreased the responsiveness of genes to muscular unloading in association with enhanced incorporation of histone variant H3.3 into nucleosomes in the plantaris muscle of rats. This indicated that endurance exercise training stimulated turnover of histones [7], but it is currently unclear how RE affects the distribution of histones and their modification patterns in animal models. Furthermore, even less research has been conducted on the epigenetic response to RE in human skeletal muscle [8-11]. Therefore, the present study built upon the animal literature, and aimed to investigate the response of gene expression profiles and histone modifications in human muscle before and after 1) acute RE and 2) chronic RE training.” in line 24 of page 3 to line 5 of page 4.

Comments:

Please indicate the static at software package used to analyze these data.

Responses:

 Statistics section was revised as “Statistical analysis was performed using BellCurve for Excel (Social Survey Research Information Co., Ltd.). For the data analysis of RNA-seq, all FPKM values were compared in same genes between all experimental groups, and calculate a correlation coefficient (R2) (Fig. 1). FPKM values were normalized using the median within each gene, and averaged in up- or down-regulated genes (Fig. 2). For ChIP data, a boxplot was used to display the distribution of the data obtained from each gene (Figs. 5 and 6). Values plotted in the figures 2, 5, 6, and 7 were compared to determine the significant differences only between pre- and post-acute RE or RE training. Because the data of pre- and post-groups were obtained from same subjects in both Experiment 1 and 2, significant differences were examined using a paired t-test. Differences were considered significant at p<0.05.” in line 21 of page 9 to line 3 of page 10.

Comments:

Paragraph 1: Add a period after this sentence: “however, are reportedly up- or downregulated

in hypertrophic or atrophic conditions in skeletal muscles of rodents [14-17].”

Responses:

 The sentence was revised as “For example, stress-responsive proteins coded by genes such as Cryab, Dnajb, Hspa1a and Hspb1 function as molecular chaperones, however, are reportedly up- or down-regulated in hypertrophic or atrophic conditions in skeletal muscles of rodents [19-22].” in the bottom line of page 12.

Comments:

Write “RE Training” throughout all figures (to be similar to “Acute AE”).

Responses:

 “Training” was reworded to “RE training” throughout all figures.

---

## [Decision Letter · Decision Letter 1]

17 Feb 2020

PONE-D-19-25531R1

Adaptive responses of histone modifications to resistance exercise in human skeletal muscle

PLOS ONE

Dear Dr Kawano,

Thank you for submitting your manuscript to PLOS ONE. After careful consideration, we feel that it has merit and both expert Reviewers provided positive support for your paper. However, a few minor things remain to be addressed which means that the paper does not fully meet PLOS ONE’s publication criteria as it currently stands yet but we believe that it will with the accompanying revisions. Therefore, we invite you to submit a revised version of the manuscript that addresses the points raised during the review process.

We would appreciate receiving your revised manuscript by Apr 02 2020 11:59PM. To enhance the reproducibility of your results, we recommend that if applicable you deposit your laboratory protocols in protocols.io, where a protocol can be assigned its own identifier (DOI) such that it can be cited independently in the future. For instructions see: http://journals.plos.org/plosone/s/submission-guidelines#loc-laboratory-protocols

We look forward to receiving your revised manuscript.

Kind regards,

Stephen E Alway, Ph.D.

Academic Editor

PLOS ONE

Reviewers' comments:

Reviewer's Responses to Questions

**Comments to the Author**

1. If the authors have adequately addressed your comments raised in a previous round of review and you feel that this manuscript is now acceptable for publication, you may indicate that here to bypass the “Comments to the Author” section, enter your conflict of interest statement in the “Confidential to Editor” section, and submit your "Accept" recommendation.

Reviewer #1: All comments have been addressed

Reviewer #2: (No Response)

2. Is the manuscript technically sound, and do the data support the conclusions?

Reviewer #1: Yes

Reviewer #2: Yes

3. Has the statistical analysis been performed appropriately and rigorously? 

Reviewer #1: Yes

Reviewer #2: Yes

4. Have the authors made all data underlying the findings in their manuscript fully available?

Reviewer #1: Yes

Reviewer #2: Yes

5. Is the manuscript presented in an intelligible fashion and written in standard English?

Reviewer #1: Yes

Reviewer #2: No

6. Review Comments to the Author

Reviewer #1: (No Response)

Reviewer #2: The authors have successfully addressed my previous comments/concerns.

However, this manuscript contains numerous grammatical errors (see examples below). I reccomend that the paper be re-reviewed for grammar by a native English speaker.

Examples:

Abstract

Remove "in the": "Exercise training causes epigenetic changes in skeletal muscle, although it is unclear how

resistance exercise (RE) affects in the histone modifications."

Should be "alters": "These results indicated that a single bout of RE drastically alter both

gene expressions and histone modifications in human skeletal muscle."

Add "is" to this sentence: " It was also suggested that enhanced histone acetylation closely related to up-regulation of gene expressions after RE training."

Discussion:

A lot of the "Limitations of analysis" section needs to be reworded to be grammatically correct. Example sentence that does not make sense: "Since scientific significance would be maximized if histone modifications were measured using the samples that was enough analyzed in histochemical characteristics, and protein expressions, the present study was motivated to analyze the remaining biopsy samples."

The section titled "Fundings" should just be "Funding".

7. PLOS authors have the option to publish the peer review history of their article (what does this mean?). If published, this will include your full peer review and any attached files.

Reviewer #1: Yes: Jarrod A Call

Reviewer #2: Yes: James R. Bagley, PhD

---

## [Author Response · Author response to Decision Letter 1]

22 Feb 2020

Responses to Reviewer 2:

Thank you for your advice for English grammar of our manuscript. Grammatical errors were corrected using a commercial English editing service. Certificate of the service is also attached to this revision.

Comment:

Remove "in the": "Exercise training causes epigenetic changes in skeletal muscle, although it is unclear how resistance exercise (RE) affects in the histone modifications."

Response:

The sentence was revised to “Exercise training causes epigenetic changes in skeletal muscle, although it is unclear how resistance exercise (RE) affects histone modifications.” in line 2-3 of page 2.

Comment:

Should be "alters": "These results indicated that a single bout of RE drastically alter both

gene expressions and histone modifications in human skeletal muscle."

Response:

The sentence was revised to “These results indicate that a single bout of RE drastically alters both gene expressions and histone modifications in human skeletal muscle.” in line 17-18 of page 2.

Comment:

Add "is" to this sentence: " It was also suggested that enhanced histone acetylation closely related to up-regulation of gene expressions after RE training."

Response:

The sentence was revised to “It is also suggested that enhanced histone acetylation is closely related to up-regulation of gene expressions after RE training.” in line 18-19 of page 2.

Comment:

A lot of the "Limitations of analysis" section needs to be reworded to be grammatically correct. Example sentence that does not make sense: "Since scientific significance would be maximized if histone modifications were measured using the samples that was enough analyzed in histochemical characteristics, and protein expressions, the present study was motivated to analyze the remaining biopsy samples."

Response:

This sentence was deleted from the Discussion.

Comment:

The section titled "Fundings" should just be "Funding".

Response:

The section title was corrected to “FUNDING” in page 17.

---

## [Decision Letter · Decision Letter 2]

23 Mar 2020

Adaptive responses of histone modifications to resistance exercise in human skeletal muscle

PONE-D-19-25531R2

Dear Dr. Kawano,

We are pleased to inform you that your manuscript has been judged scientifically suitable for publication and will be formally accepted for publication once it complies with all outstanding technical requirements.

With kind regards,

Stephen E Alway, Ph.D.

Academic Editor

PLOS ONE

Additional Editor Comments (optional):

Reviewers' comments:

Reviewer's Responses to Questions

**Comments to the Author**

1. If the authors have adequately addressed your comments raised in a previous round of review and you feel that this manuscript is now acceptable for publication, you may indicate that here to bypass the “Comments to the Author” section, enter your conflict of interest statement in the “Confidential to Editor” section, and submit your "Accept" recommendation.

Reviewer #2: All comments have been addressed

2. Is the manuscript technically sound, and do the data support the conclusions?

Reviewer #2: Yes

3. Has the statistical analysis been performed appropriately and rigorously? 

Reviewer #2: Yes

4. Have the authors made all data underlying the findings in their manuscript fully available?

Reviewer #2: Yes

5. Is the manuscript presented in an intelligible fashion and written in standard English?

Reviewer #2: Yes

6. Review Comments to the Author

Reviewer #2: (No Response)

7. PLOS authors have the option to publish the peer review history of their article (what does this mean?). If published, this will include your full peer review and any attached files.

Reviewer #2: Yes: James R. Bagley

---

## [Editor Report · Acceptance letter]

25 Mar 2020

PONE-D-19-25531R2 

Adaptive responses of histone modifications to resistance exercise in human skeletal muscle 

Dear Dr. Kawano:

I am pleased to inform you that your manuscript has been deemed suitable for publication in PLOS ONE. Congratulations! Your manuscript is now with our production department. 

With kind regards,

on behalf of

Dr. Stephen E Alway 

Academic Editor

PLOS ONE